# The Evaluation of the Effectiveness of Reinforcement by Cemented-Carbide Plates in Two Design Variants of the Chisels Intended for Cultivation–Sowing Aggregates

**DOI:** 10.3390/ma14041020

**Published:** 2021-02-21

**Authors:** Piotr Kostencki, Tomasz Stawicki, Aleksandra Królicka

**Affiliations:** 1Department of Renewable Energy Engineering, Faculty of Environmental Management and Agriculture, West Pomeranian University of Technology in Szczecin, 71-459 Szczecin, Poland; Piotr.Kostencki@zut.edu.pl (P.K.); Tomasz.Stawicki@zut.edu.pl (T.S.); 2Department of Metal Forming, Welding and Metrology, Faculty of Mechanical Engineering, Wroclaw University of Science and Technology, 50-370 Wroclaw, Poland

**Keywords:** cemented-carbide plates, abrasive wear, agricultural soil, chisels of cultivation-sowing aggregate, agricultural tools

## Abstract

Field tribological tests of two design variants of chisels used in the teeth of a cultivation-sowing unit were carried out in this research. A characteristic feature of the first variant of chisels was the reinforcement of their contact surface and almost the entire rake surface by plates made of cemented carbides. On the other hand, the second variant of chisels was reinforced only in the area of the blade by two plates made of cemented carbides, soldered on the rake face of the elements. The use of the first variant of chisels contributed to a significant reduction in the wear rate of elements, especially in terms of thickness and width loss. Effective reinforcement of the rake face, with relatively lower resistance to length reduction in the elements, raises doubts as to the validity of the use of cemented-carbide plates on almost the entire length of their rake face, because the applied variant of chisels contributed to a significantly higher price. However, the second variant of chisels effectively limited the intensity of the loss of the length of the elements, and the cause of the loss of their usefulness as part of the base material wear. It was found that the main wear mechanism of the cemented-carbide plates consisted of matrix removal under the influence of the finest fraction of the soil, which weakened the embedding of carbides, and then crushing or chipping of carbide grains from the matrix, whereas the dominant wear mechanisms of martensitic steel were grooving and micro-cutting.

## 1. Introduction

The implementation of production goals in agriculture is associated with the use of agricultural tools and machines used in cultivation and soil seasoning. Commonly, ordinary cultivation tools are used, such as plows, cultivators, and harrows, which are designed to carry out single field runs. However, the share of more specialized tools and machines, which are designed to perform several treatments at the same time, is increasing (e.g., cultivation aggregates, cultivation–sowing aggregates), which is a characteristic technology in the field of conservation crops [1,2,3]. Regardless of the adopted soil cultivation technology, the success of plant production depends to a large extent on the timeliness of agrotechnical treatment, which is determined, inter alia, by the reliability of the technical means of production. In this context, it is desirable to have a long service life of the working elements of cultivation tools and machines, regardless of whether they are single-treatment or multi-treatment tools.

The working elements of agricultural tools and machines are exposed to working environment factors during use. The main factor is the soil, the cultivation of which results in abrasive wear of these working elements. It has been highlighted in the literature that abrasive wear is one of the most common tribological processes [4], and its share in the total amount of damage to machine elements is estimated at even 50% [5,6,7]. The destructive impact of abrasive wear not only concerns the working elements of agricultural tools but also is a problem in industries such as mining, construction, or mineral processing. The wear of the working elements operating in the soil is a particularly complex tribological process. This is mainly due to the high variability of the operational forces, which are determined by the physicochemical condition of the soil and the agrotechnical parameters of the elements’ operation. Numerous scientific reports have considered the relationship between the intensity of abrasive wear and the soil moisture [8,9,10,11], the size and shape of abrasive grains [12,13,14,15], and soil pH [16,17,18]. In addition, the subject of interest of many authors has been the study of the impact of the variability of soil and agrotechnical conditions on the intensity of abrasive wear of various working elements [10,19,20,21,22,23,24,25]. In the mentioned scientific research, the authors focused on specified soil and cultivation aspects, which are not the only factors influencing the durability and reliability of the working elements operating in the soil. These parameters are also significantly determined by the material and construction factors that are used in the development of working elements.

Field studies aimed at the assessment of actual working elements when used under field conditions should be considered significant for agricultural practice. The adoption of such a method of carrying out the research forms the basis for a reliable inference about the usefulness of working elements constituting a specific material and construction solution. Of course, the results of field experiments are equally important for cognitive reasons, especially when the subject of research is the abrasive wear mechanism, which has not been fully recognized so far. Among others, Owsiak, in his research, focused on these problems, in which he proposed a mathematical model of abrasive wear for a wedge-shaped working element [26]. For the algebraic description of abrasive wear, this author used the results of a series of laboratory and field tests carried out for coulters made of four types of steel and used under variable soil conditions. Fielke performed research in which he analyzed the impact of the geometry change in the working element caused by wear in the soil on the forces acting on the element [19]. Field tests should also be considered essential for the assessment of abrasive resistance of materials used in the working elements operating in the soil. Investigations were carried out on the abrasive resistance of steels commonly used for working elements operating in the soil, depending on such material characteristics as hardness, chemical composition, and microstructure [27,28,29,30,31,32]. Research in this direction is constantly up to date due to the ongoing progress in the construction and materials of the considered working elements. New construction materials are constantly being used in agricultural applications. Currently, it is common practice to reinforce working elements by using materials characterized by higher abrasive resistance than the parent material of these elements in places particularly exposed to wear. One of the methods of improving abrasive resistance is hardfacing, the advantages of which have been confirmed in the case of cultivator coulters, plowshares, subsoiler chisels, or working elements of a rotary tiller [4,33,34,35,36,37,38,39,40]. Another increasingly used method of improving the operational parameters of working elements is their reinforcing by plates made of cemented carbides. In laboratory and field tests, the significant advantage of cemented carbides was confirmed, demonstrating that their abrasive resistance is much higher than that of steel used for working elements operating in the soil, as well as the material of padding welds [41,42,43]. However, the favorable tribological properties of cemented carbides do not guarantee a significant improvement in the durability of working elements. This may be indicated by the results of field tests of subsoiler chisels, which revealed a defect related to the brittleness of cemented carbides. During the use of chisels, large losses in the material of cemented-carbide plates were identified, which, despite its high abrasive resistance, ruled out the potential benefits of using this form of element reinforcement. The authors of the mentioned research referred to the defectiveness of the chisels with the insufficient rigidity of the base element to which the cemented-carbide plates were soldered [44]. It is worth noting that the deficiency of cemented carbides has been demonstrated in field tests, the conditions of which would be very difficult (if at all possible) to reference in laboratory tests. This may also indicate the significance of conducting this type of research under real working conditions.

This investigation aimed to evaluate two design variants of chisels used in the cultivation-sowing aggregate that were reinforced with plates made of cemented carbides by various methods of reinforcement. It was assumed that it is justified to perform field tests on working elements characterized by a significantly different design approach to reducing abrasive wear of the working elements operating in the soil. Different concepts of reinforcing correspond to a large variety of prices of the elements, which raises the question of how rational it is to use expensive materials in the construction of elements operating in the soil. In this context, the performed investigations are characterized by a utilitarian nature. However, when considering the complexity of interactions occurring in the tribological system (soil-working element), the tests performed under field conditions provide information of a cognitive nature. The results of the tests may be important due to the dominant proportion of tribological works carried out under laboratory conditions (observed in recent years) and the deficit of field tests on still improved working elements.

## 2. Materials and Methods

The field tests were carried out in 2019, during cultivation and sowing treatments, in the area of agricultural land of one of the agricultural enterprises in Poland. The subjects of the research were chisels used in the teeth of a cultivation-sowing aggregate operating in the strip-tilling system. The working width of the used aggregate was 6 m. The teeth in the aggregate were arranged in two rows. The distance between the teeth in each row was 0.6 m. The teeth were used to rip the soil into which the seeds of the cultivated plant will be sown and to introduce fertilizer into the soil. Thus, these elements are a combination of ripping and fertilizer coulters. The teeth were protected against overload (stone chipping) by spring-loaded tilting mechanisms.

Two design variants for chisels were used in this research, marked with symbols A and B (Figure 1 and Figure 2).

A characteristic feature of chisels A was the reinforcing of their contact and rake surfaces with plates made of cemented carbides. Two massive plates were soldered next to each other on the chisel blade, covering the contact surface and the rake surface of the elements. However, the function of the remaining plates was to cover the rake surface of the elements. The plates had a different thickness, which decreased with the distance from the chisel blade (except for the plates located in the area of the mounting holes) (Figure 1). The 4.5 and 3 mm thick plates were profiled according to the shape demonstrated in Figure 1.

On the other hand, chisels marked with the symbol B were reinforced only by two flat plates made of cemented carbides, located next to each other and soldered above the blade of the elements (Figure 2).

Chisels B were elements not in use, while chisels A were dismounted from the new seeder, which was tested in field cultivation at the end of the agrotechnical year preceding the research date. Macroscopic evaluation of chisels A exhibited no symptoms of abrasive wear and no mechanical damage to the cemented-carbide plates. The geometrical dimensions of all chisels A and their visual condition (visible symptoms of corrosion in the area of the base material of the elements) were similar to the condition of the selected chisel, illustrated in Figure 1.

The chemical composition of the steel as the base material of the tested chisels was determined using a GDS500A Leco glow discharge optical spectroscope (LECO Corporation, San Jose, MI, USA). However, the chemical composition of the cemented-carbide plates was performed using a JEOL JED-2300 X-ray microanalyzer (EDX, EDS, JEOL, Tokyo, Japan) coupled with a JEOL JSM-6610A scanning electron microscope (JEOL, Tokyo, Japan). The quantitative results of the chemical composition analysis obtained from the energy spectra were calculated using the ZAF correction.

Microstructure observations were carried out on the cross section of the chisels. The samples were ground with sandpaper (size in the range of 120–1200) and polished with the use of diamond pastes (diamond size: 6 µm and 1 µm). To reveal the microstructure, the samples were etched with H_2_CrO_4_ electrolyte (cemented carbides) and 3% HNO_3_ reagent (steel). Observations of the microstructure were performed using an Eclipse MA200 light microscope (LM) equipped with a Nikon DS-fi CCD and a JEOL JSM-6610A scanning electron microscope (SEM, Nikon Corporation, Tokyo, Japan). SEM observations were performed at an accelerating voltage of 20 kV and a working distance of 10 mm. The microstructure of the steel was observed using topographic contrast secondary electron (SE) detector, and the microstructure of the cemented carbides was observed using material contrast (back-scattered electron (BSE) detector, JEOL, Tokyo, Japan). Quantitative analysis of the tungsten carbide fraction in relation to the cobalt matrix (cemented carbides) was performed by graphical image editing using ImageJ software [45], with various phase contrasts (BSE detector). The threshold level was determined manually and comparably for each analyzed area. The results of the area fraction of carbide and matrix were determined based on three randomly selected areas.

Hardness measurements were made by the Vickers method in accordance with the EN ISO 6507-1: 2018 standard [46] using a Matsuzawa MMT-X7B hardness tester (steel, Matsuzawa, Akita, Japan) and a Zwick 321 hardness tester (cemented carbides, ZwickRoell, Ulm, Germany). A load of 9.81 N (HV1) and 294.2 N (HV30) was applied for 15 s. Hardness measurements were made on the cross section of the chisels. The distance between the indentations was 1 mm.

In this research, six chisels (three chisels of each type were mounted on the teeth of the first and second beams) were used. The distribution of the tested elements on the beams of the cultivation-sowing aggregate is presented in Figure 3.

The working conditions of the chisels were characterized by determining the parameters related to soil conditions (the percentage of soil species in the total area of the study, an example of the grain size of the cultivated soils, the content of gravel in the soil, the content of humus in the soil, soil reaction, soil humidity, soil volume density, soil consistency, soil shear stresses) and agrotechnical conditions (depth and speed of cultivation). The fraction of soil species was estimated based on the soil and agricultural maps of the research area. The grain size of the cultivated soils was determined for collective soil samples using the sieve-areometric method. The percentage of gravel particles was determined by sieving soil samples through a 2 mm sieve. The volumetric density and humidity of the soil were determined by the drying-weighing method using 100 cm^3^ Kopecky’s cylinders. Consistency was determined with a spring meter using a cone diameter of 11 or 12.75 mm (depending on soil conditions) with an apex angle of 30°. Shearing stresses were measured with a Geonor H-60 vane tester (Geonor Inc., Augusta, ME, USA) equipped with a cross 20 mm wide and 40 mm high or 16 mm wide and 32 mm high (also depending on soil conditions). Cultivation speed was determined by measuring the cultivator travel time with a stopwatch at a distance of 50 m.

The change in the geometry of the chisels, resulting from the wear impact of the soil, was determined by measuring the absolute thickness, width, and length of the elements. The dimensions of the elements were measured before and after their use in the soil. The measurement points are shown in Figure 4. The selection of measurement points was to a large extent determined by the shape of the chisels and the used measuring equipment: The thickness loss of the elements was measured with a micrometer with an accuracy of ±0.001 mm; the width loss was measured by a caliper with an accuracy of ±0.05 mm: while to determine the length loss, the outlines of elements were made, and on this basis, measurements were made with an accuracy of ±0.5 mm. The limitation in the choice of measurement points concerned, in particular, the thickness measurement in the initial area of the elements due to the geometry of the elements in this area.

It should be added that for chisels A, measurement points g_2_ to g_8_ and measurement lines b_2_ to b_8_ were located in the soldering zone of the cemented-carbide plates, while for chisels B, these points corresponded to the area of the base material of the elements. In addition, measurement point g_9_ and measurement line b_9_ for chisels A and B were located in the area of their base material.

Based on the performed linear measurements, a change in the geometry of the chisels was determined in the indicated measurement points. The results of these measurements were used to designate the unit value of the thickness, width, and length loss of chisels, which was determined in relation to the friction path of the elements. The friction path of the chisels was determined based on the cultivated area and the working width of the cultivation–sowing aggregate.

The topography of the worn chisel surfaces in selected zones was also observed using scanning electron microscopy (JEOL JSM-6610A, detector SE, topographic contrast). Based on the worn surface topography, the wear mechanisms of the chisel material were determined and identified. The surface zones of the chisels that were heavily loaded by the soil were selected for the SEM observation. The selected zones in which the observations were carried out are presented in Figure 4.

## 3. Results and Discussion

### 3.1. Working Conditions of Chisels

During the research, the cultivation–sowing aggregate worked at sowing rape seeds in the fields after wheat or barley harvest. Table 1 demonstrates the values of parameters describing the working conditions of chisels, and Table 2 illustrates the results of measurements of an exemplary grain size distribution of the sown soils.

Due to the reached wear limit, chisels B were dismounted earlier than chisels A. Thus, their working conditions were slightly different. Chisels B, compared to chisels A, worked in slightly heavier soil, with slightly higher humidity, lower consistency, and lower shear stresses. Other parameters characterizing the soil and agrotechnical working conditions of the elements were comparable (Table 1).

In the case of chisels A, the friction path of the elements was 628.2 km, and in the case of B chisels, 258.7 km (this corresponded to a cultivation acreage of 376.9 ha for chisels A and 155.2 ha for chisels B; in the case of chisels A, it was the total area of the field intended for the cultivation of rape in this agrotechnical season). The reason for the earlier disassembly of chisels B was a large decrease in their thickness and width; moreover, one of the elements mounted on the first row broke at the place of the lower assembly hole (Figure 5). Chisels A were characterized by slight wear before the disassembling of chisels B. For this reason, chisels A were operated for a longer time, which made it possible to determine the geometry of the elements with a greater degree of their wear. In this way, it was possible to precisely determine the values of the parameters describing the wear intensity of materials used in the construction of these elements.

### 3.2. Characterization of Materials Used in the Construction of Chisels

The chemical composition and hardness of the materials used in chisels A and B are presented in Table 3, and their microstructure is shown in Figure 6, Figure 7 and Figure 8.

In both chisel variants, medium-carbon, low-alloy boron steels were used as the base material (Table 3). The microstructure of the steel used in chisels A consisted of martensite and ferrite arranged in bands (Figure 6c,d). The presence of ferrite in martensitic steels increases the impact toughness and fracture toughness, while martensite determines the abrasive wear resistance. In contrast, the base material of chisels B exhibited a tempering martensite microstructure (Figure 7c,d). The measured hardness of the base material for chisels A (435.6 ± 14.3 HV1) was almost 100 HV units lower than that for chisels B (529.0 ± 5.4 HV1) (Table 3). The lower hardness of the steel used in chisels A resulted directly from the presence of ferrite in their microstructure.

The cemented-carbide plates for both chisel variants were composed of tungsten carbide (WC) grains and a cobalt matrix. For chisels B, the content of the cobalt matrix was slightly higher than that for chisels A (Table 3). On the other hand, based on graphical image analysis, it was determined that the cobalt matrix share in chisels A and chisels B was 19.04% ± 0.65% and 20.56% ± 53%, respectively. Examples of the images after the threshold process used for the calculations are presented in Figure 8. Moreover, for both variants of chisels, WC grains differed in refinement (Figure 6a,b and Figure 7a,b, respectively). Based on the measurements of the grain size of carbides and in accordance with the PN-EN ISO 4499-2: 2008 standard [47], chisels A were classified as coarse (2.5–6.0 μm) and in chisels B as fine (0.8–1.3 μm). Generally, the cemented-carbide plates were characterized by a comparable level of hardness (chisels A, 1133 ± 28 HV30; chisels B, 1029 ± 27 HV30). The slightly lower hardness of the plates of chisels B could be caused by a higher fraction of the cobalt matrix compared to chisels A.

The cemented-carbide plates were joined to the base material of the chisels by a high-temperature brazing process (Figure 9) using Cu-Ni-based filler materials. The thickness of the solder in chisels B was about four times greater than that in chisels A. No macro- and microscopic solder incompatibilities were identified.

### 3.3. Unit Loss of the Thickness, Width, and Length of Chisels

Figure 10, Figure 11 and Figure 12 demonstrate the unit values of the loss of thickness, width, and length of chisels A and B determined in the measurement points. The impact of the location of the chisels in the tractor tracks on the intensity of their wear was not identified. Thus, the parameters given in Figure 10, Figure 11 and Figure 12 are average values for all chisels of a given type, divided into the results obtained for the elements operating in the first and second beams.

The value of the unit thickness loss of chisels B decreased with the distance from the blade of the elements (Figure 10), except for measurement point g_2_, where the parameter value was lower than at point g_3_ (taking into account the chisels of the first and second beams, approximately 1.9 times). The lower value of the unit thickness loss occurring at the g_2_ measurement point was probably related to its location: the g_2_ point was located close to the cemented-carbide plates (Figure 4), which covered the base material in this area.

In the case of chisels B, it is noticeable that the value of the unit loss of their thickness decreased with the distance from the blade of the elements (the lower value of the parameter at measurement point g_2_ can be attributed to the protective effect of the cemented-carbide plates fixed in the area of the blade) (Figure 10). The value of the unit loss was also lower in the case of elements mounted on the second beam. This dependence concerned all measurement points, with the notable exception of point g_7_, where an opposite tendency was noticed. At the measuring points g_2_, g_3_, and g_4_, the value of the unit ratio of the loss of the thickness of chisels from the first and second beams ranged from 2.05 to 2.33; at points g_5_, g_6_, and g_8_, from 1.12 to 1.58; at point g_9_, 1.01; and at point g_7_, 0.66. This may indicate that the chisels of the first beam were heavily loaded by the soil, in particular in the area close to the blade of the elements.

In the case of chisels A, the highest value of the unit thickness loss was also found in the initial area of the elements: measurement points g_1_ and g_2_ (excluding large values of the parameter occurring at point g_9_, where the base material of the chisels was not covered with plates made of cemented carbide) (Figure 10). However, the trend of a decreasing value of the unit thickness loss with the distance from the chisels’ blade was not as clear as it was in the case of chisels B. Attention should be paid to the generally high values of the standard deviation for the unit value of the chisel thickness loss occurring at individual measurement points. This proves the significant influence of random factors on the wear process of tested elements. It appears that the wear of the chisels was influenced by slight differences in the structure and properties of the materials used in tested elements, as well as the heterogeneity of the cultivated soil.

In the case of chisels A, it is also noticeable that the value of the unit loss of their thickness was lower in the case of the elements mounted on the second beam; such a situation was identified for six of nine measurement points (Figure 9).

A large unit loss of the width of chisels B occurred in measurement lines b_1_, b_2_, and b_3_, i.e., near the blade of the elements (Figure 11). The slightly lower value of the parameter in line b_1_ can be explained by the protective effect of the plates made of cemented carbide located in this line (Figure 4). It should be emphasized that the large values of the unit width loss of the elements occurring in lines b_2_ and b_3_ corresponded to the large values of the unit thickness loss of the chisels occurring in points g_2_ and g_3_ (Figure 10 and Figure 11).

In the range of measurement lines b_4_ to b_9_, the value of the unit width loss of chisels B successively decreased. This may indicate a lower soil load for the higher area of the tested elements.

In the case of chisels A, the values of the unit width loss occurring in given measurement lines were much less differentiated than in the case of chisels B (Figure 11). Again, the relatively large values of the standard deviation for the unit width loss of chisels A should be mentioned, which indicates a significant impact of random factors on the wear process of the tested elements. On the other hand, low and even values of the unit width loss prove a protective effect of cemented-carbide plates also in terms of reducing the width of the elements. A noticeable tendency should be highlighted, i.e., the occurrence of lower values of the unit width loss of chisels A for thicker plates made of cemented carbide. Measurement lines b_1_ and b_2_ were located in the area of 4.5-mm-thick plates, lines b_3_ to b_6_ in the area of 3-mm-thick plates, and lines b_7_ and b_8_ in the area of 1-mm-thick plates. Overall, the lowest values of the unit width loss of chisels A occurred in lines b_1_ and b_2_ and the largest in lines b_7_ and b_8_ (Figure 11): the average unit width loss of lines b_7_ and b_8_ was about four and two times greater than in lines b_1_ and b_2_ for chisels mounted on the first and second beams, respectively.

The measurement results of the unit width loss of chisels A and B also indicate, although less unambiguously, that the working conditions of the elements mounted on the first beam are heavier. A higher value of the unit width loss of chisels located in the first beam as compared to the wear of chisels located in the second beam was found in five measurement lines for both tested chisels A and B.

The unit length loss of chisels B, measured along the axis of the element (measurement line l_1_), was comparable for the elements mounted on the first and second beams (Figure 12). With unused chisels B, the base material of the elements protruded about 2 mm in front of the soldered cemented-carbide plates (Figure 2). For this reason, for these chisels, the values of the unit length loss of the entire elements (with the material of the blade not covered by the cemented-carbide plates) were determined, additionally taking into account only the reduction in the size of the plates made of cemented carbide. The values of the unit length loss of chisels B determined based on the change in the dimension of the cemented-carbide plates were about 15% lower than the parameter values determined based on the total loss of the length of the elements.

In the case of chisels A, the unit length loss of the elements mounted on the first beam was about 1.2 times higher than for the elements mounted on the second beam, which again confirms that the working conditions in the first beam were more severe.

### 3.4. Wear Mechanism of Materials Used in Tested Chisels

Figure 13, Figure 14, Figure 15 and Figure 16 demonstrate the wear mechanisms of chisel materials occurring in the zones indicated in Figure 4. In the wear processes of cemented-carbide plates, in both chisel variants tested, the matrix was initially removed under the influence of the finest fractions of abrasive mass. Cracks and matrix defects appeared around the carbides, which weakened the carbide embedding in the cobalt matrix (Figure 13, Figure 14, Figure 15 and Figure 16). In the second stage of wear, the weakened carbides, under the influence of abrasive working conditions, cracked or chipped out of the matrix, resulting in the formation of characteristic craters (pits) with a shape corresponding to the removed carbide grains (Figure 13b, Figure 14b and Figure 15b). It was also found that some of the carbides firmly embedded in the matrix were chipping or cracking (Figure 13b and Figure 14a). The process of chipping and cracking of carbides was less pronounced in the case of finer WC grains that occurred in chisels B (Figure 15b). In the case of fine-grained cemented carbides, the WC grains weakened and poorly embedded in the partially removed matrix were usually chipped out. Grinding effects were identified in chisels B (Figure 15a,b). A similar effect, but less severe in the cobalt matrix area, was present in the Z1 area of chisels A (Figure 13b). In the case of the cemented-carbide plates used in chisels A, in the zone of presumably maximum loads (area Z1), intensified carbide chipping and the formation of discontinuities and pits were observed (Figure 13) compared to the less loaded area Z2 (Figure 14).

However, in zone Z4 of chisels B (Figure 4), martensitic steel was worn. The dominant wear mechanisms of steel were grooving and micro-cutting (Figure 16). The observed scratches related to the micro-cutting mechanism were characterized by a smaller width and greater depth compared to grooves (Figure 16b). In general, the direction of the grooves and scratches was consistent with the movement of the abrasive fraction; only a low fraction of them differed in the direction. The edges of the grooves were characterized by the presence of slight local plastic deformation (Figure 16b). The presence of pinholes was also identified, which may indicate the impact of large particles hitting the surface of the base material of the tested chisel (Figure 16).

The wear mechanisms of cemented-carbide plates, as well as martensitic steel, indicate the abrasive and erosive nature of wear. The erosion was caused by the dynamic impact of the larger fraction on the surface of the chisels.

### 3.5. Comparison of Tested Chisels in Terms of Wear Resistance

In the case of a unit loss of the thickness, width, and length of the tested chisels, significantly lower values of these parameters were found for chisels A, i.e., elements reinforced with a series of plates made of cemented carbide. Table 4 presents the ratio values of the unit losses in the thickness, width, and length of chisels B in relation to the wear of chisels A. These ratios were calculated for individual measuring areas where the change in element geometry was determined. The ratio values clearly illustrate the greater wear resistance of chisels A. It is noticeable that large ratio values of the unit thickness and width loss occur for the initial area of the elements (Table 4), which was related to the intensive wear of chisels B in this area (Figure 10 and Figure 11). On the other hand, these multiplicities decreased as a function of the distance from the blade of the elements, while in only one measurement point, the wear of chisels A was greater than the wear of chisels B (the decrease in the width of the elements at measurement point b_7_, elements operating in the second beam; Figure 11 and Table 4). The highest value of the ratio of the thickness loss of chisels B and A occurred at measurement point g_3_, while the lowest occurred at point g_9_, where the base material of chisels B and A was subject to wear.

Measurement line b_2_ was the zone where the highest value of the ratio of the width loss in chisels B and A occurred. For chisels A, this measurement line included a cemented-carbide plate (4.5 mm thick), while for chisels B, there was only the base material. The low wear in the width of chisels A in this line corresponded to a relatively high loss of the width of chisels B (Figure 11), which resulted in a large value of the described ratio.

The multiplicity of the length loss in chisels B in relation to the length loss in chisels A was not as high as it was for the thickness or width of the tested elements (Table 4). Thus, the use of cemented-carbide plates significantly contributed to reducing the shortening rate of both chisels. Nevertheless, chisels B were worn about five times faster than chisels A. This can be explained by the application of cemented-carbide plates in chisels A, which covered both the rake and the contact surfaces of the chisels and were much more massive than the flat plates used in the construction of chisels B (Figure 1 and Figure 2). It should be noted again that in chisels B, the plates made of cemented carbide were soldered at a distance of about 2 mm from the blade of the elements (Figure 2). Thus, during the use of the elements, the base material was worn first and then the cemented-carbide plates. Taking into account the wear of only cemented-carbide plates for chisels A and B, the ratio of the unit length loss in chisels B to the length loss in chisels A was slightly lower and amounted to approximately 4.4.

Earlier, a comparison of the tested design variants of chisels was performed based on the value of unit loss of their material, which should be considered as one of the possible criteria for evaluating elements. From the user’s point of view, the agrotechnical quality of soil cultivation is also important, which is conditioned by changes in the geometry of the elements resulting from their wear. Another important factor that determines the rational use of a given design variant of working elements operating in the soil is their price referred to the cultivated field area until the elements reach the wear limit state.

For both chisels A and B, the use of cemented-carbide plates reduced the destructive impact of the soil on the elements. The application of cemented-carbide plates in chisels B had a positive effect on the reinforcing of their blades. As a consequence, the length loss in chisels B did not determine their durability. The loss of usability of chisels B should be associated with the intensive wear of the base material occurring above the plates made of cemented carbide, which led to a large decrease in the thickness and width of the elements in this area (Figure 5). Therefore, it can be concluded that the blade of chisels B were reinforced by plates made of cemented carbides and that such reinforcement has an advantageous effect on maintaining the cultivation depth. Similar observations can be made with regard to chisels A, the blade of which was also covered with plates made of cemented carbide. In the case of these elements, additionally, almost the entire rake surface was reinforced (Figure 1), which effectively limited the intensity of the loss of their thickness and width. These elements were not exposed to fracture due to a large loss of material above the blade area, as was the case with chisels B (Figure 5).

The market price of chisels A was about 6.7 times higher than the price of chisels B, while the acreage of the field cultivated at the time of the research was about 2.4 times greater than that of chisels B. However, during the tests, chisels B were subjected to wear close to the wear limit (chisels were exposed to fracture as a result of a large thickness and width loss). On the other hand, chisels A disassembled after field tests were characterized by relatively low wear. However, the obtained results of measurements allowed us to determine the values of the parameters of the unit material loss of the elements, based on which the possible further wear process of chisels A can be determined. In terms of the thickness loss of cemented-carbide plates, the highest wear intensity was found for chisels A mounted on the first beam, at the g_2_ point: 0.556 mm per 1000 km of friction path. The initial thickness of these plates was 4.5 mm, assuming a constant rate of their wear, giving a utility potential of about 8000 km of friction path (which corresponds to 4800 ha of the field cultivated by a cultivation-sowing aggregate). The correspondingly estimated utility potential of 3 and 1 mm thick plates is approximately 8600 km and 3850 km, respectively. The presented analysis demonstrates exquisite protection of chisels A against the intense rate of thickness loss. The cemented-carbide plates also effectively protected chisels A against changing their width in the area of the elements most loaded by the soil. This is indicated by the low unit width loss in chisels A in measurement lines b1 to b4 (located close to the blade; Figure 11). In the analyzed cross sections, the unit change in the width of chisels A compared to chisels B was many times lower: from about 22.5 times in cross-section b3 up to 133.5 times in cross-section b2 (Table 4). The values given apply to elements mounted on the first beam and, therefore, are subject to greater wear than the elements mounted on the second beam.

However, the measurement results indicate that the usefulness of chisels A is determined to a greater extent by the change in their length than by the loss of thickness and width. In the case of chisels A mounted on the first beam of the cultivation-sowing aggregate, the unit length loss was 6.1 mm per 1000 km of friction path, which, for example, is a value almost 11 times greater than the previously mentioned value of the unit thickness loss of the cemented-carbide plates. The value of the parameter was estimated based on the length loss of the elements after performing 628.2 km of the friction path when the blades of the elements were still covered from the side of the rake and contact face with properly profiled plates made of cemented carbides (Figure 1). During further use of the chisels, the plates would wear, where the blade of the elements would no longer be protected from the contact surface side. The wear dynamics of a partially protected blade would change, and thus the value of the unit loss of the length of elements would increase. Therefore, it can be assumed that the price of chisels A, which was 7.6 times higher than the price of chisels B, would not be compensated by their longer service life resulting from the use of a large number of cemented-carbide plates and covering the rake surfaces of the chisels with them.

## 4. Conclusions

The use of reinforcement by soldering cemented-carbide plates on their blade and almost the entire rake surface (chisels A) compared to the wear of chisels that were reinforced with cemented-carbide plates only in the area of the blade (chisels B) contributed to a significant reduction in the wear rate of elements, especially in terms of the loss of their thickness and width.

Effective reinforcing of the rake face of chisels A, with a relatively low resistance to the shortening of elements, questions the validity of using cemented-carbide plates over almost the entire length of their rake surface, because the applied variant of reinforcement of chisels A contributed to a much higher price in relation to the price of chisels B. Nevertheless, chisels A enable longer operation of the cultivation–sowing aggregate without replacing the working elements.The reinforcement of the blade of chisels B by cemented-carbide plates reduced the intensity of loss of length. In the context of the service life loss of the elements as a result of the wear of their base material not protected by plates, such a design variant should be considered an effective form of chisel reinforcement.There were different wear mechanisms of the used materials. In the case of plates made of cemented carbide, the matrix was removed under the influence of the finest fraction of soil, which weakened the embedding of carbide grains in the cobalt matrix and then their chipping or cracking. The process of chipping and cracking carbides was less intensive in the case of fine WC grains (chisels B). The grinding effect was also identified in the area of the highest soil loads on the elements, which was more pronounced in chisels B (fine WC grains). On the other hand, the dominant wear mechanisms of martensitic steel used in chisels B were grooving and micro-cutting. Overall, the nature of wear was both abrasive and erosive.The wear rate of the chisels installed in the first beams was higher, which indicates a greater load on their working surfaces from the soil. This dependence, especially in terms of thickness and width loss, was found for chisels B (reinforced only in the area of the blade of the chisels).

## Figures and Tables

**Figure 1 materials-14-01020-f001:**
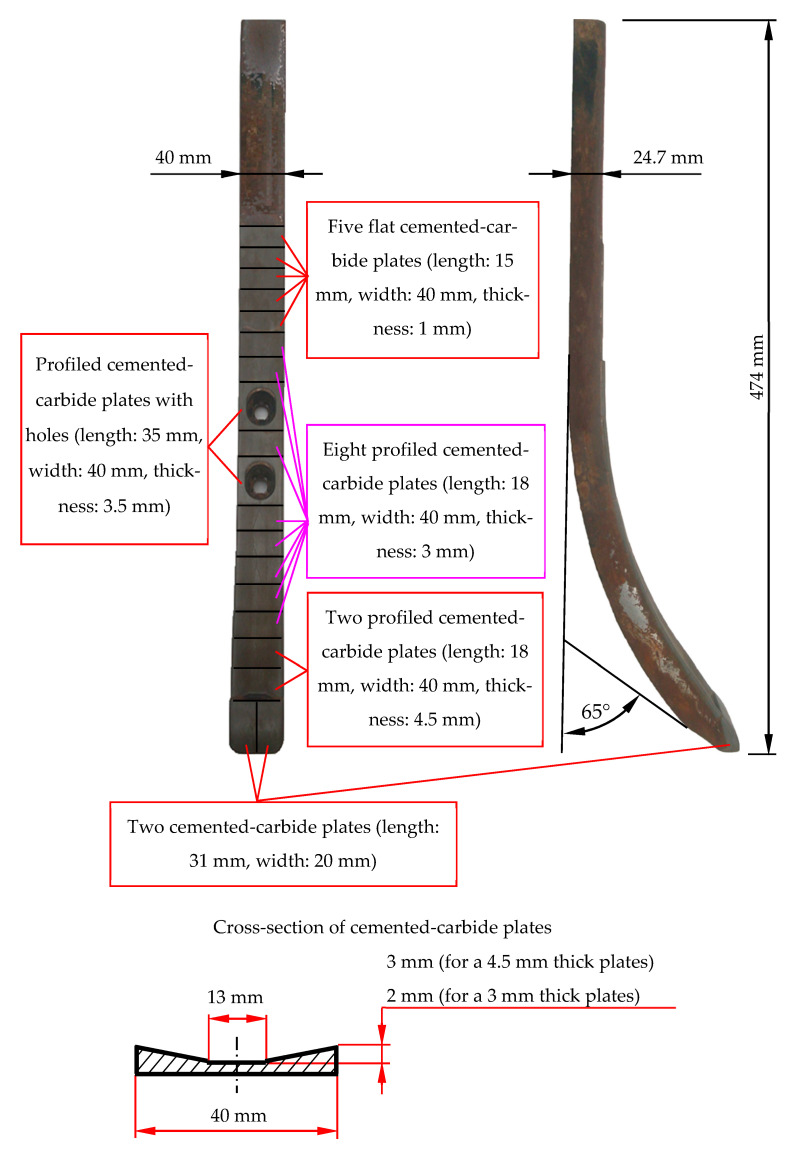
Tested chisels: variant A.

**Figure 2 materials-14-01020-f002:**
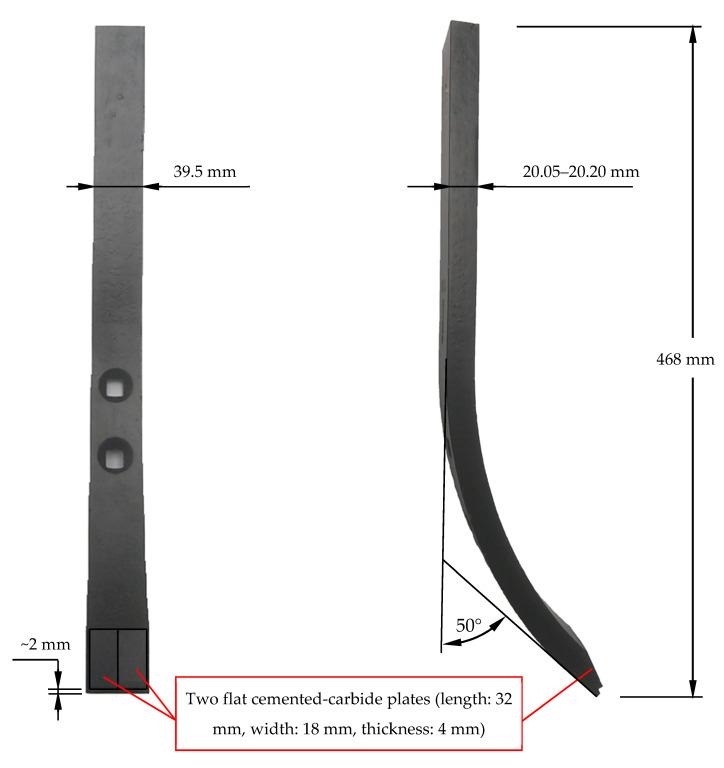
Tested chisels: variant B.

**Figure 3 materials-14-01020-f003:**
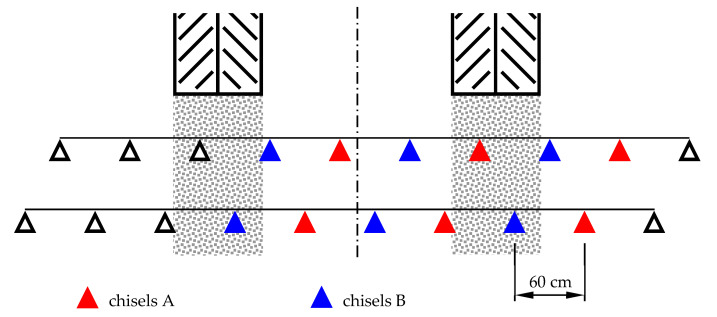
Distribution of tested elements on beams in a cultivation–sowing aggregate.

**Figure 4 materials-14-01020-f004:**
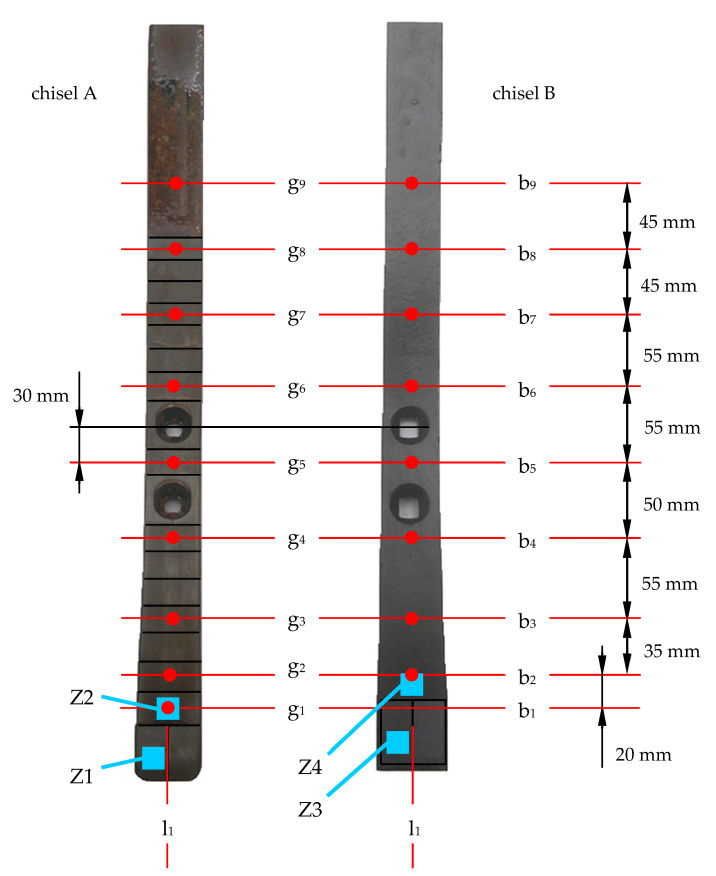
Thickness loss measurement points (for chisels A from g_1_ to g_9_, for chisels B from g_2_ to g_9_), width loss measurement lines (from b_1_ to b_9_), length loss measurements line (line l_1_), and scanning electron microscope (SEM) observation zones (from Z1 to Z4) of the rake surface topography of chisels after their operating in the soil.

**Figure 5 materials-14-01020-f005:**
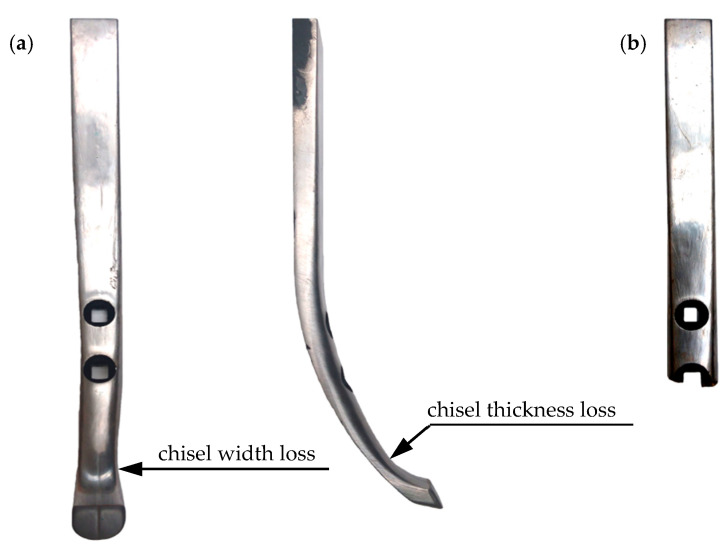
The geometry of the selected chisel B after disassembly: (**a**) element without the wear limit state and (**b**) broken element.

**Figure 6 materials-14-01020-f006:**
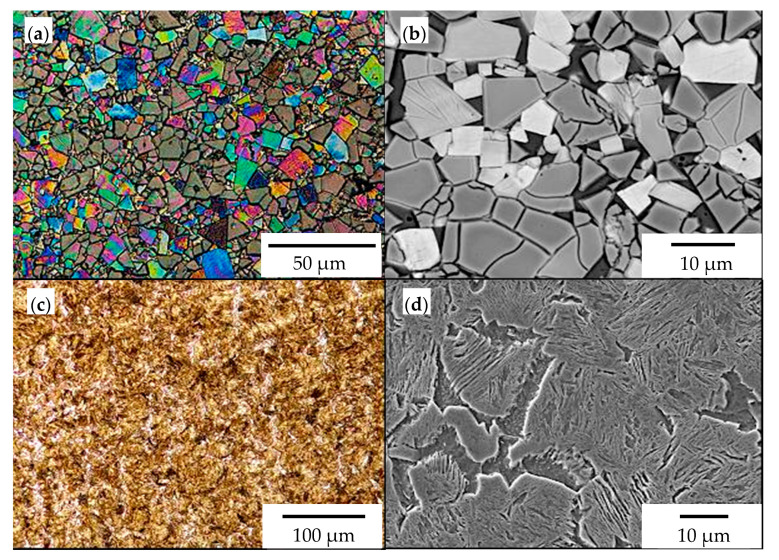
The microstructure of materials used for chisels A: (**a**) cemented-carbide plate (light microscope (LM)), (**b**) visible coarse grains of tungsten carbide in a cobalt matrix (SEM, back-scattered electron (BSE) detector), (**c**) microstructure of the base material (LM), and (**d**) visible medium-carbon martensite and ferrite (SEM, secondary electron (SE) detector).

**Figure 7 materials-14-01020-f007:**
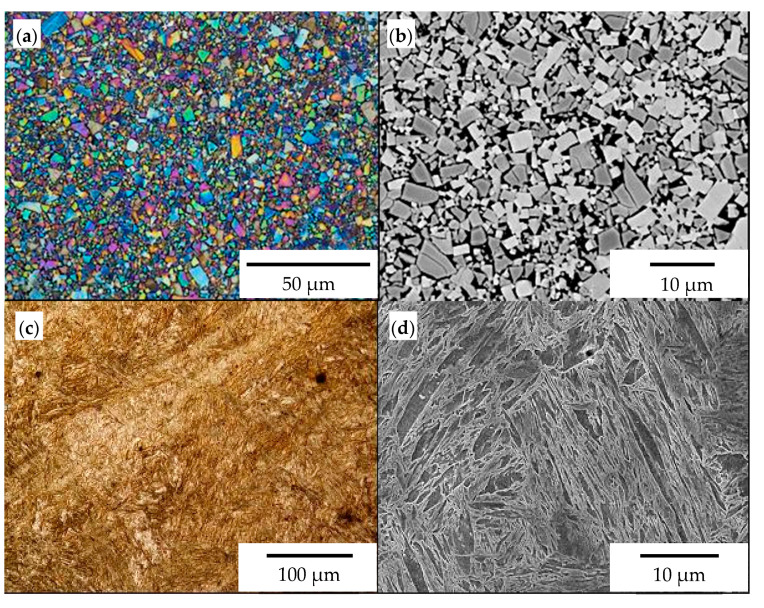
The microstructure of materials used for chisels B: (**a**) cemented-carbide plate (LM), (**b**) visible fine grains of tungsten carbide in a cobalt matrix (SEM, BSE detector), (**c**) microstructure of the base material (LM), (**d**) visible medium-carbon tempered martensite (SEM, SE detector).

**Figure 8 materials-14-01020-f008:**
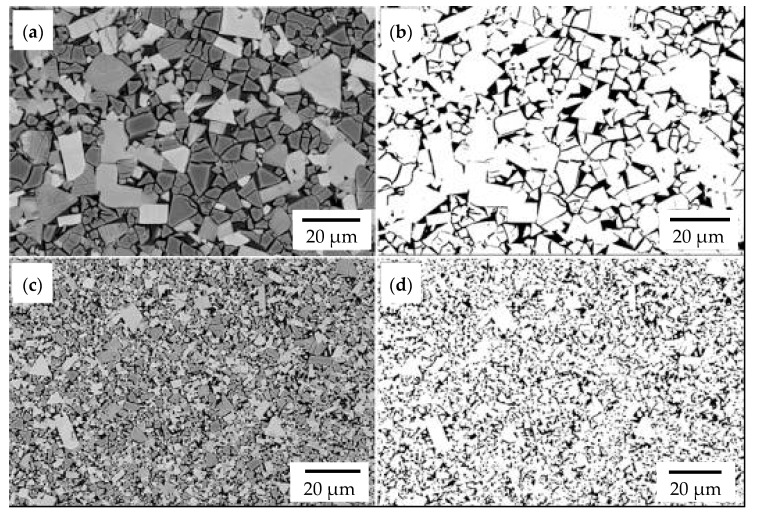
An example of the areas used for quantitative graphical image analysis. Reference images of chisels A (**a**) and chisels B (**c**). Images after threshold intended for calculation of carbide and matrix share of chisels A (**b**) and chisels B (**d**). SEM, BSE detector.

**Figure 9 materials-14-01020-f009:**
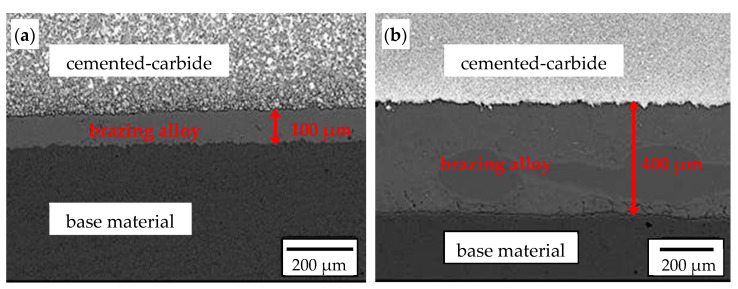
Cross section of the brazed joint of the cemented-carbide plate with the base material: (**a**) chisels A and (**b**) chisels B. SEM, BSE detector.

**Figure 10 materials-14-01020-f010:**
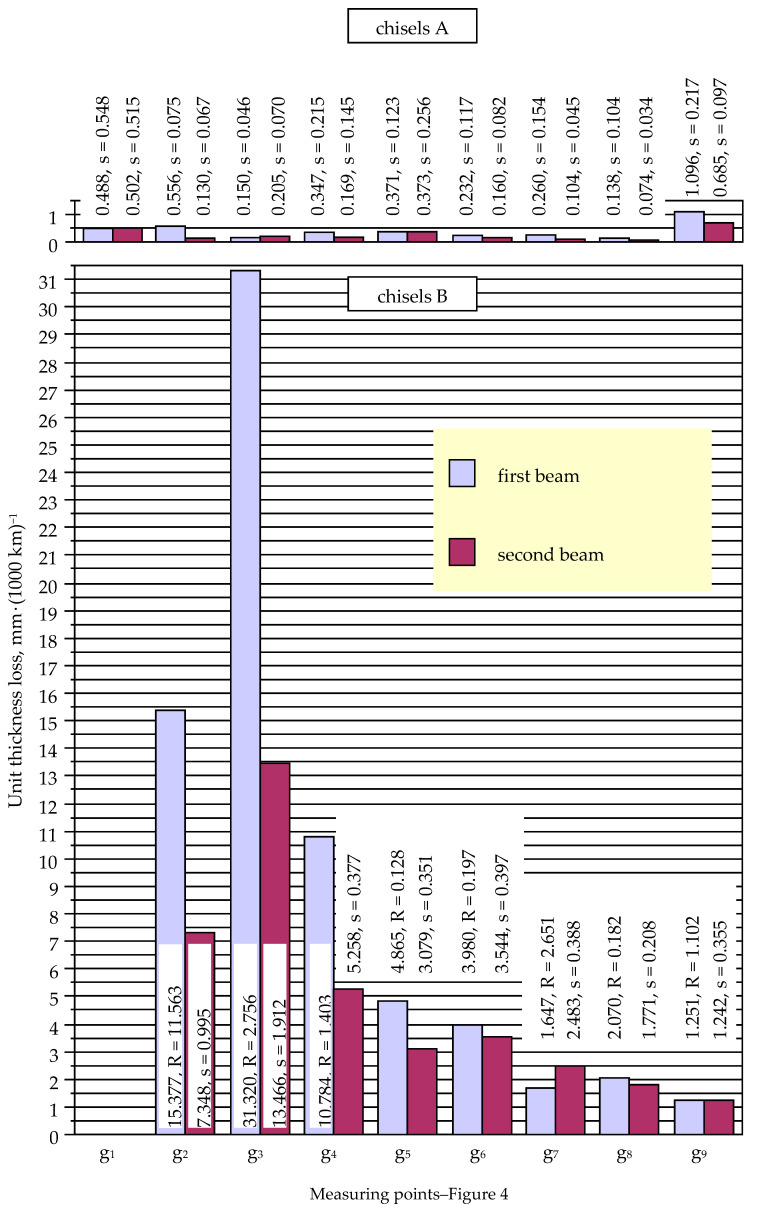
Unit thickness loss of chisels A and B (s, standard deviation; R, range).

**Figure 11 materials-14-01020-f011:**
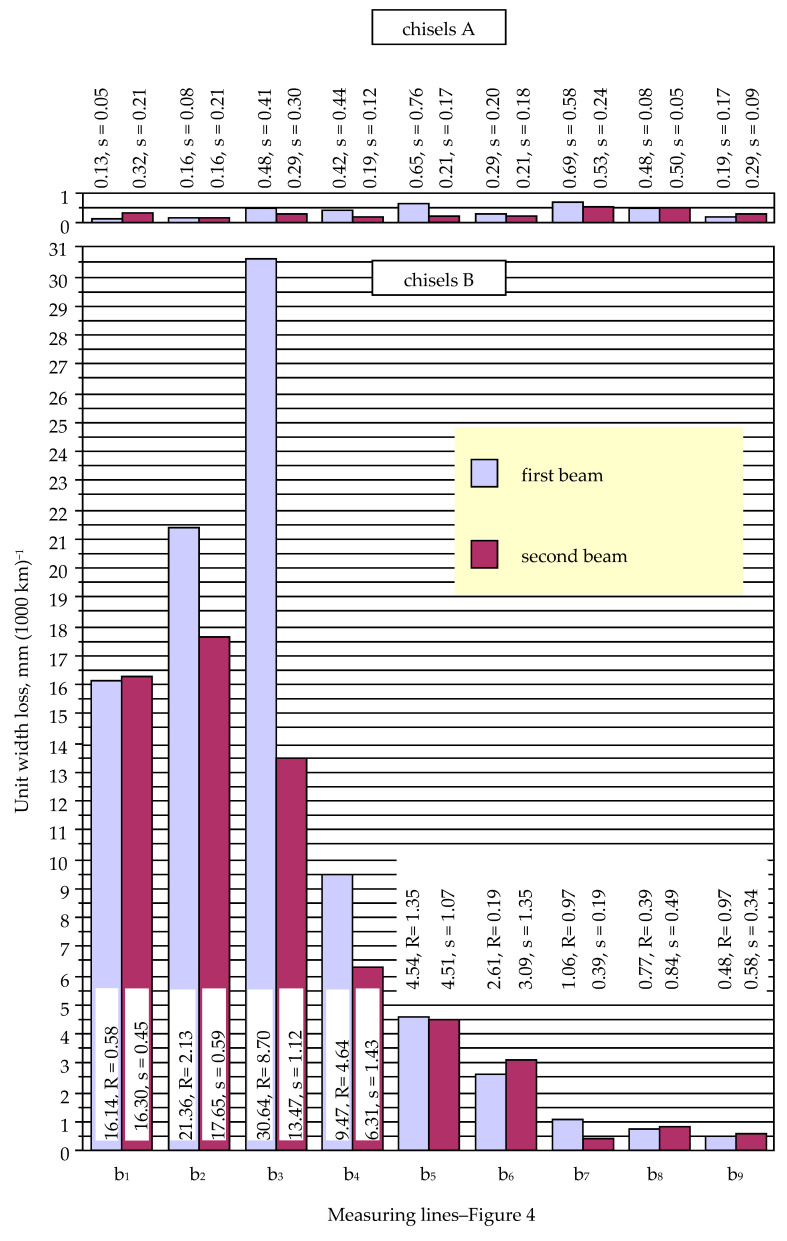
Unit width loss of chisels A and B (s, standard deviation; R, range).

**Figure 12 materials-14-01020-f012:**
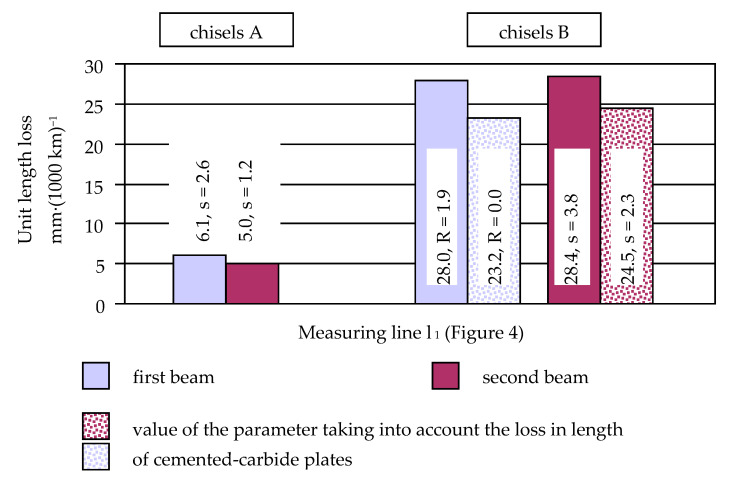
Unit length loss of chisels A and B (s, standard deviation; R, range).

**Figure 13 materials-14-01020-f013:**
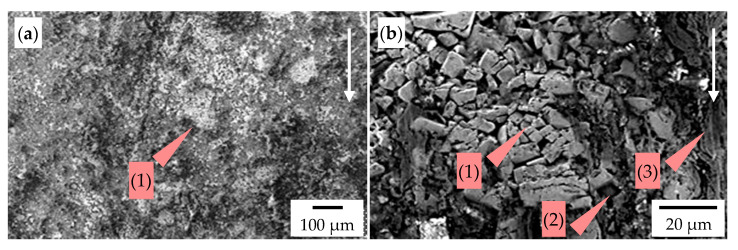
The worn surface of the cemented-carbide plate operating in the soil: chisel A, zone Z1 indicated in Figure 5; the white arrow presents the direction of the impact of soil particles. (**a**) Visible pits (1) caused by chipping out of tungsten carbide (WC) grains from the cobalt matrix and (**b**) visible crushed WC grains (1), pits (2), and traces of the interaction of soil particles in the cobalt matrix (3). SEM, SE detector.

**Figure 14 materials-14-01020-f014:**
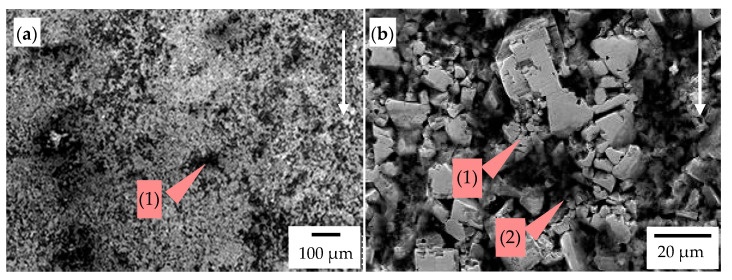
The worn surface of the cemented-carbide plate operating in the soil: chisel A, zone Z2 indicated in Figure 5; the white arrow presents the direction of the impact of soil particles. (**a**) Visible pits (1) caused by chipping out of WC grains from the cobalt matrix and (**b**) visible crushed WC grains (1) and pits (2). SEM, SE detector.

**Figure 15 materials-14-01020-f015:**
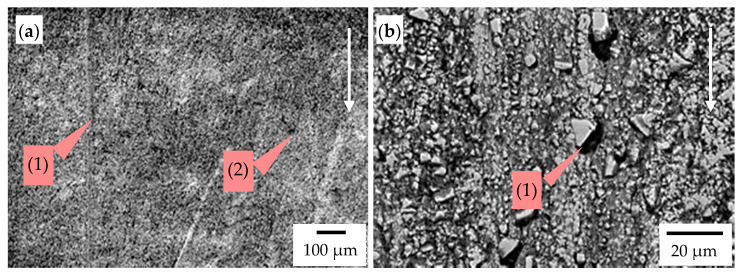
The worn surface of the cemented-carbide plate operating in the soil: chisel B, zone Z3 indicated in Figure 5; the white arrow presents the direction of the impact of soil. (**a**) Visible traces of the interaction of soil particles in the cobalt matrix (grinding effect) (1) and pits (2) and (**b**) visible WC grains poorly embedded in the cobalt matrix (1), which was removed. SEM, SE detector.

**Figure 16 materials-14-01020-f016:**
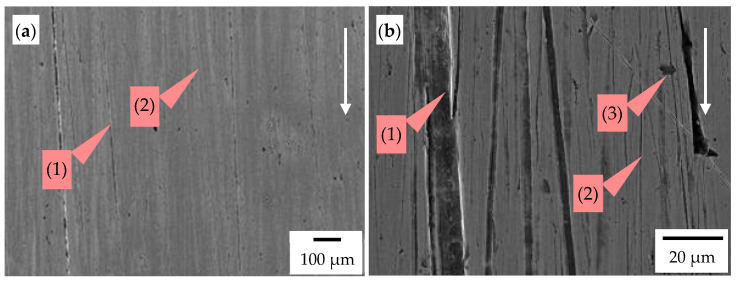
The worn surface of the base material (steel) operating in the soil: chisel B, zone Z4 indicated in Figure 5; the white arrow presents the direction of the impact of soil. (**a**) Visible grooves (1) and scratches (2) oriented following the movement of soil particles and (**b**) visible grooves (1), scratches (2), and pinholes (3). SEM, SE detector.

**Table 1 materials-14-01020-t001:** Working conditions of examined elements.

Quantity	Soil Layer	Parameter Value			
		Chisels A		Chisels B	
percentage of soil granulometric groups in the research area, %	Arable layer	Sandy loam	26	59	
		Loamy sand	20	26	
		Light loamy sand	48	10	
		Sand	6	5	
percentage of gravel (2–30 mm), %		4.6, s = 3.0			
percentage of humus, %		1.76, s = 0.44			
reaction, pH_KCl_		5.40–7.43			
actual humidity, wt%	0–15 cm	7.7	s = 1.9	8.5	s = 1.7
	15–30 cm	6.4	s = 1.1	7.1	s = 0.7
volumetric density, g⋅cm^−3^	0–15 cm	1.46	s = 0.06	1.43	s = 0.05
	15–30 cm	1.43	s = 0.05	1.42	s = 0.07
consistency, kPa	0–15 cm	2150	s = 1029	1829	s = 364
	15–30 cm	3369	s = 1069	3197	s = 1352
shearing stress, kPa	0–15 cm	79	s = 34	55	s = 8
	15–30 cm	111	s = 37	93	s = 40
working depth, cm		28.1	s = 2.2	28.0	s = 2.7
working speed, m⋅s^−1^		2.76	s = 0.11	2.74	s = 0.12

**Table 2 materials-14-01020-t002:** Exemplary percentages of granulometric fractions in the cultivated soil.

G	Percentage of Granulometric Fraction, %							Granulometric Group
	Sand					Silt 0.002 < d ≤ 0.05	Clay d ≤ 0.002	
	Very Coarse 1.0 < d ≤ 2.0	Coarse 0.5 < d ≤ 1.0	Medium 0.25 < d ≤ 0.5	Fine 0.10 < d ≤ 0.25	Very Fine 0.05 < d ≤ 0.10			
1	2.1	5.4	13.1	28.5	20.6	26.4	3.9	FSL
2	1.9	5.0	12.3	25.1	16.4	31.5	7.8	FSL
3	5.2	8.6	16.9	32.1	7.8	25.6	3.8	FSL
4	2.3	7.1	13.4	27.3	17.7	28.3	3.9	FSL
5	2.0	5.1	12.0	23.4	16.3	36.3	4.9	FSL

d, the size of soil grains, mm; FSL, fine sandy loam.

**Table 3 materials-14-01020-t003:** Chemical composition and hardness of materials used in chisels.

Chisel	Material	Chemical Composition, wt%	Hardness
A	Base material	0.306C-1.200Mn-0.245Si-0.103Cr-0.270Al-0.014P-0.001S-0.034Ti-0.002B	435.6 ± 14.3 HV1
	Cemented-carbide plates	Tungsten carbide (WC), 85.99 ^(1)^; Co matrix, 14.01 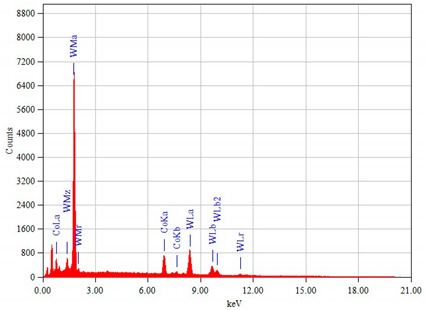	1133 ± 28 HV30
B	Base material	0.250C-1.270Mn-0.254Si-0.369Cr-0.210Al--0.010P-0.007S-0.034Ti-0.001B	529.0 ± 5.4 HV1
	Cemented-carbide plates	Tungsten carbide (WC), 83.28 ^(1)^; Co matrix, 16.72 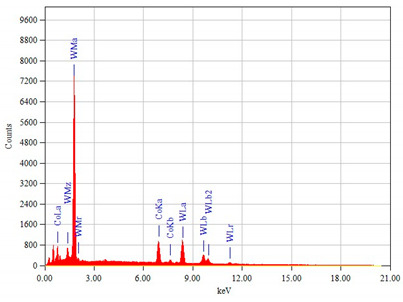	1029 ± 27 HV30

^(1)^ The content of tungsten carbide was determined based on the tungsten content, assuming the atomic ratio of carbon and tungsten.

**Table 4 materials-14-01020-t004:** The ratio of the unit loss of thickness, width, and length of chisels B in relation to the wear of chisels A.

The Multiplicity of Unit Thickness Loss of Chisels B in Relation to the Wear of Chisels A									
beam	Measurement point								
	g_2_		g_3_	g_4_	g_5_	g_6_	g_7_	g_8_	g_9_
first	27.7		208.8	31.1	13.1	17.2	6.3	15.0	1.1
second	56.5		65.7	31.1	8.3	22.2	23.9	23.9	1.8
beam	line								
	b_1_	b_2_	b_3_	b_4_	b_5_	b_6_	b_7_	b_8_	b_9_
first	124.2	133.5	63.8	22.5	7.0	9.0	1.5	1.6	2.5
second	50.9	110.3	46.4	33.2	21.5	14.7	0.7	1.7	2.0
beam	Measurement line								
	l_1_								
first	4.6, (3.8) *								
second	5.7, (4.9) *								

*** Taking into account the length loss only of the cemented-carbide plates.

## Data Availability

Data available on request due to restrictions of privacy. The data presented in this study are available on request from the corresponding author.

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
