# Peer review of "The Evaluation of the Effectiveness of Reinforcement by Cemented-Carbide Plates in Two Design Variants of the Chisels Intended for Cultivation–Sowing Aggregates"

_materials, 2021, doi:10.3390/ma14041020_

Round 1

Reviewer 1 Report

  1. Pay attention to the unit superscript. For example, cm3 (line 178).
  2. Lost the length unit in line 183 (32).
  3. Pay attention to the consistency of proper noun abbreviations. For example, WC carbide, tungsten (WC), tungsten carbide (WC).
  4. It is the best that the experimental data results can be presented in a quantitative improvement rate.
  5. The abstract and conclusion can be more concise and important result statement.

Author Response

Dear Reviewer, 

please find our responses in the attachment.

Yours sincerely
Aleksandra Królicka

Reviewer 2 Report

  1. The introduction is sufficient and provides a good background on the relevant research publication. However, the added value and motivation of this work are not strongly highlighted.
  2. In the experimental part you should mention the number of repeats per test.
  3. Description of how the coatings were produced should be added in the text. 
  4. Wear mechanism: You do not only have abrasion, but also erosion due to the impacting of hard particles from the soil (as you correctly identified from SEM analysis).
  5. Is the change of the chisel angle due to wear or due to bulk deformation (from the torque during motion)?
  6. EDS analysis is very localized and cannot be used to quantify carbon.
  7. On the characterization of the coatings you should mention the Vol% of the carbide particles.
  8. You mention that random factors have an influence on wear. Do you refer to structural characteristics, coating features, soil repeatability etc. This will also help you to justify the unrepeatable nature of wear.  
  9. It would have been nice if you focused more on the correlation between the structure of the coatings and their wear performance. This journal is about Materials, so addressing this will increase the impact of your article.           

Author Response

(The authors gave the same response as above.)

Round 2

Reviewer 2 Report

After reading the updated version of the manuscript and your point-by-point reply to reviewers, I know believe that this article is suitable for publication.